# A Continental-Scale Assessment of Density, Size, Distribution and Historical Trends of Farm Dams Using Deep Learning Convolutional Neural Networks

**Martino E. Malerba [1,*], Nicholas Wright [2] and Peter I. Macreadie [1]**

1 Centre for Integrative Ecology, School of Life and Environmental Sciences, Deakin University, Melbourne, VIC 3125, Australia; p.macreadie@deakin.edu.au
2 Sustainability and Biosecurity, Department of Primary Industries and Regional Development, 3 Baron-Hay Court, South Perth, WA 6151, Australia; Nicholas.Wright@dpird.wa.gov.au
* Correspondence: m.malerba@deakin.edu.au

**Abstract:** Farm dams are a ubiquitous limnological feature of agricultural landscapes worldwide. While their primary function is to capture and store water, they also have disproportionally large effects on biodiversity and biogeochemical cycling, with important relevance to several Sustainable Development Goals (SDGs). However, the abundance and distribution of farm dams is unknown in most parts of the world. Therefore, we used artificial intelligence and remote sensing data to address this critical global information gap. Specifically, we trained a deep learning convolutional neural network (CNN) on high-definition satellite images to detect farm dams and carry out the first continental-scale assessment on density, distribution and historical trends. We found that in Australia there are 1.765 million farm dams that occupy an area larger than Rhode Island (4678 km$^2$) and store over 20 times more water than Sydney Harbour (10,990 GL). The State of New South Wales recorded the highest number of farm dams (654,983; 37% of the total) and Victoria the highest overall density (1.73 dams km$^{-2}$). We also estimated that 202,119 farm dams (11.5%) remain omitted from any maps, especially in South Australia, Western Australia and the Northern Territory. Three decades of historical records revealed an ongoing decrease in the construction rate of farm dams, from >3% per annum before 2000, to ~1% after 2000, to <0.05% after 2010—except in the Australian Capital Territory where rates have remained relatively high. We also found systematic trends in construction design: farm dams built in 2015 are on average 50% larger in surface area and contain 66% more water than those built in 1989. To facilitate sharing information on sustainable farm dam management with authorities, scientists, managers and local communities, we developed AusDams.org—a free interactive portal to visualise and generate statistics on the physical, environmental and ecological impacts of farm dams.

**Keywords:** water security; artificial water bodies; GIS; deep learning classification models; Australian management policies; land-use change; surface water detection

## 1. Introduction

Farm dams are a ubiquitous feature of agricultural landscapes and a cornerstone of farming and industrial practices. These artificial water bodies are relatively small ($10^2$–$10^5$ m$^2$) and collect water for livestock and irrigation, protect against fires and more [1,2]. Escalating water prices, diminishing rainfalls and increasing temperatures are stimulating the development of new farm dams, increasing worldwide at up to 60% per annum to respond to increasing pressure on agricultural production [3]. While often ignored from inventories, recent assessments showed that small (100–1000 m$^2$) aquatic ecosystems dominate the areal extent of continental waters worldwide with substantial contributions to global cycles [3].

It is increasingly more apparent that the accumulation of farm dams can have intensifying effects on biodiversity [4,5], nutrient cycling [6], soil erosion [7], biological

invasions [8] and other biogenic processes. Recently, farm dams have also been recognised as significant sources of greenhouse gas to the atmosphere, ranking among the highest emitters per unit area among freshwater systems [9,10]. As a result, the 2019 refinement of IPCC Guidelines recommends accounting for farm dams in national carbon inventories. Hence, developing "greener" solutions for managing farm dams can have essential contributions to many of the Sustainable Development Goals (SDGs) recently identified by the United Nations, including reduce climate change, preserve biodiversity and increasing water security. However, many countries lack the most basic information on farm dams, such as their density and location, hindering our understanding of how to manage their environmental and ecological effects [11].

Remote sensing is a cost-effective approach to map surface water at large scales. Deep learning convolutional neural networks (CNN) are among the most advanced algorithms for image analysis and their popularity in remote sensing is rising rapidly [12]. By extracting features at different hierarchical levels, CNN have tremendous potential to improve accuracy, precision and generalisation to detect water in satellite images over alternative machine learning techniques [12–14]. For example, CNN algorithms could accurately identify water bodies despite the presence of clouds, ice, snow, glare and shadows [12–15]. However, the computer-intensive nature of CNN still represents a substantial challenge. Indeed, most previous implementations for surface water detection remain either at low definition (e.g., Landsat or Sentinel data with 10–30 m spatial resolution in [12,14–16]) or small/medium geographic scales (e.g., urban areas in [13,17]).

Australia is a large and dry country, covering 5.6% of the world's landmass but only containing 1% of its total freshwater [18,19]. Water is, therefore, a limited resource and a critical policy concern [20]. In Australia, there has never been a formal assessment of farm dam densities or total counts, with Federal authorities and scientific articles reporting ballpark estimates from "half a million" [4], to "over two million" [21,22], to "several million" [23]. The Australian Government (i.e., Geoscience Australia) has previously invested in a Water Observation from Space program [24]. However, the minimum detection is limited to water bodies larger than half a soccer field (50 × 50 m), which excludes the majority of farm dams. The only nation-wide dataset of Australian farm dams is by Geoscience Australia [25,26], but only a subset of dams features in this map. For example, the number of farm dams reported by Geoscience Australia for the State of Tasmania (N = 726) is 1% of those expected by local authorities (N = 61,897—see Table S1).

We present the first continental-scale assessment of density, distribution, water capacity and historical trends of farm dams. First, we compiled all available information from Federal, State and local authorities on Australian farm dams and similar artificial water bodies (e.g., irrigation ponds, sewage ponds, settling ponds—see Table S1). Second, we used CNN to detect farm dams from satellite images to account for different levels of uncertainty among data sources. Third, we analysed historical trends in the rate of development of new farm dams from each State and Territory in Australia.

## 2. Materials and Methods

### 2.1. Mapping Farm Dams in Australia

Please refer to Table S1 for details on curator, spatial coverage, temporal coverage, sample size, data type, filters, access date and source for all datasets used in this study. Briefly, we sourced data on 1,694,675 farm dams from (1) the Surface Water map by Geoscience Australia (N = 934,381), (2) the Department of Environment, Land, Water & Planning of the Victorian Government (N = 429,398), (3) the Department for Environment and Water in South Australia (N = 105,361), (4) the Department of Primary Industries and Regional Development in Western Australia (N = 162,785), (5) the Department of Primary Industries, Parks, Water and Environment in Tasmania (N = 61,897) and (6) the Environment & Planning Directorate in the Australian Capital Territory (N = 853). For large farm dams (>$10^5$ m$^2$ in surface area), we removed those that appeared of natural origins (i.e., complex shapes, jiggered borders) by retaining only those with simple and regular shapes, calculated

as circularity ($4 \times Area \times \left[\pi \times Perimeter^2\right]^{-1}$) above 0.5. For farm dams reported as points (as opposed to polygons), we used the minimum detection area for polygons noted in the metadata, and we calculated the perimeter assuming circular shape. We ensured there were no repeated or overlapping entries in our data. Finally, we developed a calibration curve to estimate water capacity (unit: ML) from the surface area (unit: m$^2$) by compiling data for 558 farm dams across Victoria, Queensland and South Australia [27–29] (model: $\log_{10}$(Water Capacity) = $-3.593$ [$-3.707$; $-3.479$] + 1.237 [1.204; 1.270] $\times \log_{10}$(Surface Area); R$^2$ = 0.91; F$_{1,556}$ = 5359.6, *p* < 0.001; see Figure 1).

*2.2. Quantifying Uncertainty*

To account for different uncertainties among jurisdictions, we developed an independent water detection algorithm that we could use to benchmark each map. Specifically, we used our algorithm as ground truth, and we derived statistical models to estimate the probabilities of false positive (i.e., entries that are wrongly classified as a dam, or commission error) and false negative (i.e., farm dams that failed to be identified, or omission error) in each State and Territory in Australia.

2.2.1. Water Detection Using Deep Learning Convolutional Neural Networks

We trained a deep learning convolutional neural network (CNN) to detect farm dams using the Python-based open-source library "fastai" version 1 https://github.com/fastai/fastai; [30]. We downloaded the most recent (typically between 2018 to 2019) RGB satellite image of 7362 Australian locations from three different repositories (i.e., http://ecn.t3.tiles.virtualearth.net, https://api.mapbox.com and https://server.arcgisonline.com). We sampled 75% of these images from our dam dataset and the remaining 25% from randomly selected locations within Australia. These satellite images had varying sizes and aspect ratios, and the pixel resolution was mostly 0.45 m, but when unavailable, we also used lower resolutions (e.g., 1–5 m).

To avoid manual labelling of all 7362 downloaded images, we took a random subsample of 400 images and labelled them into "dam" or "not dam" and we trained a classification model on the labelled data. We utilised transfer learning by initialising an ImageNet pretrained ResNet34 model [30]. We applied an 80–20% split for training and validation datasets, respectively. To help generalise the model, we used data augmentation with the fastai get_transforms function [30] and the following arguments: "flip_vert = TRUE" to allow for vertical flipping of images, "max_lighting = 0.02" to limit overly exposing the images, "max_zoom = 1" to disable the zooming augmentation, and "to_fp16 = TRUE" to reduce the memory load on the graphical processing unit (GPU). We set the batch size to 300 images and trained the model with a learning rate of $10^{-3}$ for ten epochs. At epoch 5, we achieved an error rate of 0.1538 (15.38%) a validation loss of 0.4211 and a training loss of 0.8287. We used the trained model to automatise the classification of 500 more images from the unlabelled training dataset, and we manually fixed any mistakes. We repeated this process of training, classification and checking until all of the 7362 downloaded images were labelled.

We trained our deep learning CNN on 7362 labelled images using the same parameters detailed above, and we achieved an error rate of 0.1195 (11.95%) with a training loss of 0.3462 and a validation loss of 0.2847. We further fine-tuned the model by unfreezing the entire model and training at a 10-fold lower learning rate ($10^{-4}$). The final model achieved an accuracy of 94.8% (error rate of 5.2%) with a training loss of 0.1397 and a validation loss of 0.1446 with ten epochs (see confusion matrix in Figure S1).

2.2.2. Correcting for False Positives

Locations falsely classified as containing a dam (i.e., false positives, or commission error) act to overpredict the real number of dams in Australia. Therefore, we calculated the probability of false positives by using our deep learning CNN to analyse and validate ca. 2000 dams in each State and Territory sampled from our compiled database. To do so,



we downloaded RGB satellite imagery for each farm dam using the same three repositories mentioned above and combined their predictions to generate an outcome for each location (either "dam correctly verified" or "dam being a false positive").

We corrected our dataset for false positives using generalised linear models. We used the classification outcome from our deep learning CNN (binomial distribution) as the response variable, and we used the State or Territory identity (categorical), dam surface area (continuous), and their interaction as the covariates in the analysis. The rationale is that jurisdictions using outdated or low-definition satellite images to map water bodies will have a higher probability of miss-recording smaller dams (i.e., low reliability). We used our best-fitting statistical model (following Akaike information criterion [31]) to predict the reliability (i.e., probability of true positives) for each dam in our dataset. Finally, we corrected our data by removing all entries that recorded less than 75% reliability of being a true positive, which we verified to be an appropriate threshold to filter out the large majority of false positives.

### 2.2.3. Correcting for False Negatives

Undocumented farm dams (i.e., false negatives, or omission error) underestimate the real number of dams in Australia. We estimated the fraction of undocumented dams in each State and Territory by conducting an independent exploration using our deep learning CNN to analyse areas supposedly free of dams—following our compiled map of Australian farm dams (see Table S1). Given that the probability of encountering a dam by randomly sampling a site across the whole of Australia is very low, we maximised our sampling efforts by selecting only land types with high dam densities. To do so, we overlapped our compiled dataset of Australian dams with the 2016 Australian Land Use and Management Classification (version 8) [32] to identify the 33 land types with the highest dam densities (>2 dams km$^{-2}$) in Australia (see Figure S2 for the list of these land types). Then, we randomly sampled locations in each State and Territory and downloaded RGB images (mostly from 2018 to 2019) at 0.5 m resolution from http://ecn.t3.tiles.virtualearth. net/. The number of investigated sites depended on the available sampling area at each combination of land use type by State/Territory and was typically between 11,000 to 30,000—although we could only sample fewer sites in the Northern Territory (N = 5400) and the Australian Capital Territory (N = 62). In total, we analysed 124,510 RGB images across Australia and detected 5,105 farm dams (see Figure S3 for few examples of the farm dams identified by the deep learning CNN). We calculated the relative density of false negatives compared to the density of true positives to calculate a probability of false negatives per area for each land-use type in each State and Territory. Finally, we used the mean probability of false negatives across all land-use types to estimate the total number of undocumented dams in each State and Territory.

As an example, suppose there are 100 dams documented for a specific land use type of 100 km$^2$ in size (i.e., reported density of 1 dam per km$^2$), from which five dams are removed because deemed false positives (i.e., less than 75% reliability of being a true positive; see Section 2.2.2). Hence, the farm dam density for this hypothetical land use after correcting for false positives is (100–5/100 = ) 0.95 dam per km$^2$. Were we to find 1 undocumented dam by searching 10 km$^2$ of randomly sampled locations, we would infer a density of undocumented dams of (1/10 =) 0.1 dam per km$^2$. In this case we would conclude that undocumented dams in this land use type are (0.1/0.95 =) 10.5% of the documented dams. By repeating these operations across the 33 land-use types, we could calculate an overall percentage of false negatives relative to true positives, which we used to estimate the overall number of documented + undocumented dams in each State or Territory. Our approach assumes that the probability of a dam being undocumented is constant across all land use types. This assumption is reasonable when the same mapping technique is used across the landscape, which is the case for all maps used in this study (see Table S1). Finally, we manually traced the surface area of 221 randomly selected unreported dams to estimate the median surface area (m$^2$) of undocumented dams in each

State and Territory (Figure S2), which we used to estimate the total surface area and water content in documented + undocumented dams.

### 2.2.4. Compounding Multiple Uncertainties

We quantified the overall uncertainty for all our metrics using bootstrapping procedures [33,34]. Specifically, we created 1000 simulated datasets by sampling observations with replacement. For each simulated dataset, we repeated the steps detailed above to calculate all statistics for documented + undocumented dams, including the uncertainty in estimating the water capacity of a farm dam from its surface area (see Figure 1). Finally, we extracted the median and the 95% confidence intervals from the obtained bootstrap distribution of each estimate.

### 2.3. Historical Trends

Please refer to Figure S4 for a step-by-step graphical diagram of this analysis. We used data from the Water Observations from Space (WOfS) to quantify historical changes in surface water in Australia from 1988 to present [24]. The WOfS uses Landsat 5 and Landsat 7 satellite images to detect surface water at a 30 m grid size across Australia at an approximate bi-weekly frequency. The Digital Earth Australia Waterbodies elaborates data from WOfS to provide 28 years of bi-weekly time series of relative wet surface area for 300,000 waterbodies across Australia. First, we filtered for farm dams by extracting the water bodies that overlapped with our farm dam database (see Section 2.1). Among the overlapping farm dams, we randomly selected ca. 1000 from each State and Territory—excluding the Northern Territory that had too few documented dams. For each selected farm dam, we used data from WOfS to compile bi-weekly time series of the relative number of pixels inside the farm dam area that were identified as water from 1988 to 2015. Then, we recorded the year when the WOfS time series started to consistently report water in at least 25% of the farm dam area, which was taken as the year when the farm dam was created. Finally, we calculated the relative and absolute cumulative distribution of farm dams over time in each State and Territory and used linear models to analyse historical trends (see Figure S4 for graphical diagram of the methods).

### 2.4. Statistical Analyses

We used Python [35] and fastai [30] for developing the deep learning CNN. We used R [36] for all statistical analyses, using packages sf [37] and raster [38] for data manipulation; ggplot2 [39], rasterVis [40] and cowplot [41] for plotting. We also used R for designing the website AusDams.org, using Shiny [42], Leaflet [43], Plotly [44] and using Joe Cheng's Superzip template (https://shiny.rstudio.com/gallery/superzip-example.html).

## 3. Results

### 3.1. Reported Farm Dams

There were 1,694,675 farm dams reported by regional and Federal authorities in Australia. The majority of farm dams were in New South Wales (37%), Victoria (26%), Queensland (17%) and Western Australia (10%; Table S1). Around three-quarters of Australia recorded at least one dam per 2000 $km^2$, but the typical density near urban centres was 2–5 farm dams per $km^2$ (Figure 2). The average size of a dam was ca. 1000 $m^2$, ranging from 100 $m^2$ to >$10^5$ $m^2$ (Figure 3).

### 3.2. Data Verification

Our results showed that reports of larger (>1000 $m^2$) farm dams were reliable, with a probability of a successful verification ranging from $78 \pm 1.2\%$ in Queensland to $93 \pm 1.5\%$ (S.E.) in Western Australia (Figure 3). Instead, reports of smaller farm dams (<100 $m^2$) were only verified in $34 \pm 13\%$ (in Northern Territory) to $74 \pm 13\%$ S.E. (in Western Australia) of cases (Figure 3 and Figure S5). Overall, we corrected for false positives in the data by removing 43,295 farm dams (2.55% of the total), ranging from 54 (2.5%) in the Australian

Capital Territory to 22,949 (5.18%) in Victoria. The State with the largest percentage of false positives was Tasmania (13%, Figure 3 and Figure S5). Notice that in the Northern Territory there were too few documented farm dams to carry out a formal probability assessment of false positives, so we assumed 100% of the 2040 documented dams were successfully verified.

### 3.3. Undetected Farm Dams

We estimated that 202,119 farm dams remain undetected in Australia. Farm dams in Queensland, New South Wales, Victoria and the Australian Capital Territory contributed to 80% of all documented farm dams and recorded the lowest percentages (<7%) of undetected dams, which corresponded to 94,266 omitted farm dams across the four regions (Figure 4). We recorded higher percentages of undetected farm dams in South Australia (11%), Tasmania (19%) and Western Australia (28%), for an estimated 94,108 omitted farm dams (Figure 4). Finally, the Northern Territory recorded the highest percentage (87%) of omitted farm dams (Figure 4).

### 3.4. Total Farm Dams in Australia

Overall, we estimated that in 2018/2019 there were 1,765,152 farm dams (95% C.I.: 1,668,319 to 1,907,440) in Australia. New South Wales recorded the highest number of farm dams (654,983, 37% of the total) and Victoria the highest overall density (1.73 dams km$^{-2}$; Figure 5A,B). Conversely, the Australian Capital Territory recorded the lowest dam counts (2144, 0.01% of the total) and the Northern Territory the lowest dam density (0.0026 dams km$^{-2}$). In total, farm dams in Australia occupied an area of 4678 km$^2$ (95% C.I.: 4388 to 5245).

In all regions, the large majority (>89%) of farm dams were documented (see green bars in Figure 5D), except in Tasmania (81%), Western Australia (72%) and in the Northern Territory (13%; see red bars in Figure 5D). Finally, false positives were generally a small fraction (<5%) of the total number of documented dams, with only Tasmania recording a relatively high value (10%; see blue bars in Figure 5D).

### 3.5. Total Water Stored in Dams

We estimated that the total water stored in Australian farm dams was 10,990 GL (95% C.I.: 9,434 to 13,473; Figure 5C). New South Wales recorded the greatest amount of water stored in farm dams (4,266 GL, 38.8% of the total), followed by Queensland (2720 GL, 25%) and Western Australia (1,555 GL, 14%; Figure 5C). Overall, undetected farm dams stored 1,372 GL of water (12.5% of the total). Importantly, water from undetected farm dams contributed to 95.9% (59 GL) of the total stored water in the Northern Territory, 31% (189 GL) in South Australia and 24% (372 GL) in Western Australia.

### 3.6. Historical Trends

The years between 1988 and 2000 recorded the fastest increases in farm dam numbers across all regions (>2% per annum; see steep lines in Figure 6 before the vertical dashed line). In these years, the Australian Capital Territory recorded the fastest rate of growth (3.2% per annum), and New South Wales registered the highest number of new farm dams built per year (13,948; Figure 6 and Figure S6). These rates correspond to farm dams doubling in number every 13 to 32 years.

After 2000, the development of new farm dams slowed down across Australia to <1.2% per annum, except in the Australian Capital Territory where rates remained relatively high (2.5% per annum; see lines in Figure 6 after vertical dashed line). After 2000, Queensland recorded the highest number of new farm dams built each year (3710), followed by New South Wales (2338) and Victoria (2181; see Figure S6 for all absolute and relative rates). However, we also detected a significant increase in farm dam size over time (Figure S7). Specifically, farm dams built in 2015 were on average 50.7% larger than those built in 1988—regardless of the State or Territory (i.e., no sign. interaction between region and year

of construction). This increase in surface area corresponded to a 66% increase in water capacity (Figure S7).

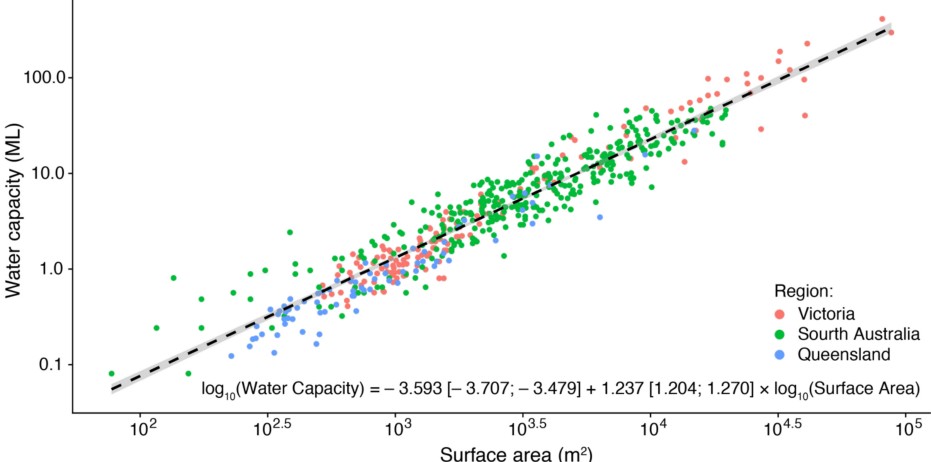

**Figure 1.** Calibration curve and model coefficients [±95% confidence intervals] to estimate the water capacity of a farm dam from its surface area ($R^2 = 0.91$, $F_{1,556} = 5359.6$, $p < 0.001$). We compiled data from three Australian studies (N = 558): farm dams in South Australia were sourced from McMurray (2004), those in Queensland from Sinclair Knight Merz (2012) and those in Victoria from Lowe et al. (2005). Water capacity was calculated using GIS techniques and Light Detection And Ranging (LIDAR) data. Surface area was calculated from satellite images. The range of surface areas covered in this relationship (from $10^2$ to $10^5$) is representative of the full range found among Australian farm dams (cf. Figure 3). The bootstrapping methods to estimate the statistics on total water capacity in Australian farm dams (Figure 5C) included the uncertainty in this relationship.

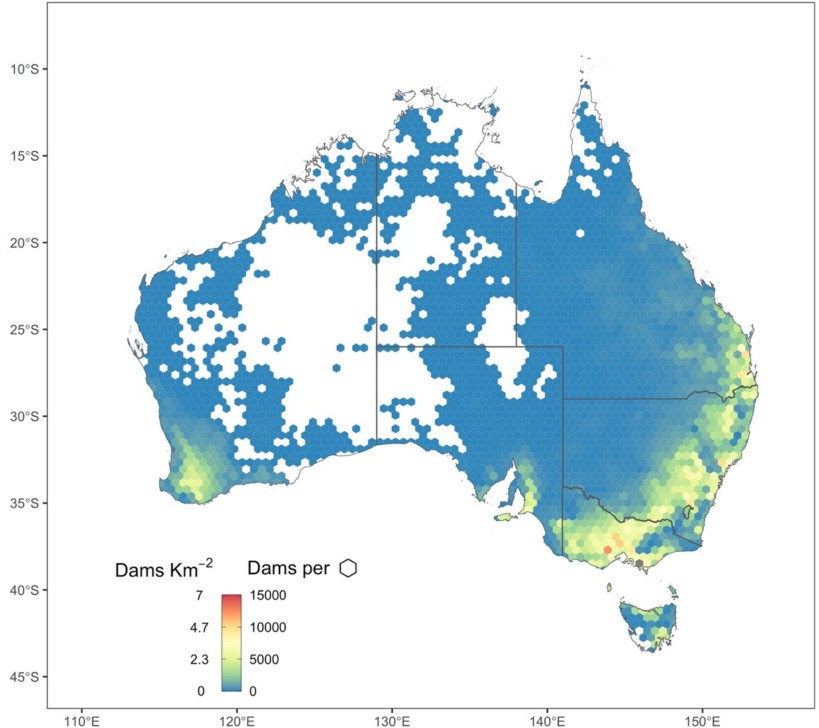

**Figure 2.** Distribution of documented dams in each Australian State and Territory, compiled from the maps in Table S1. The colour represents density (dams km$^{-2}$) and total counts (dams per hexagon), with empty hexagons indicating no reports of dams in the area.

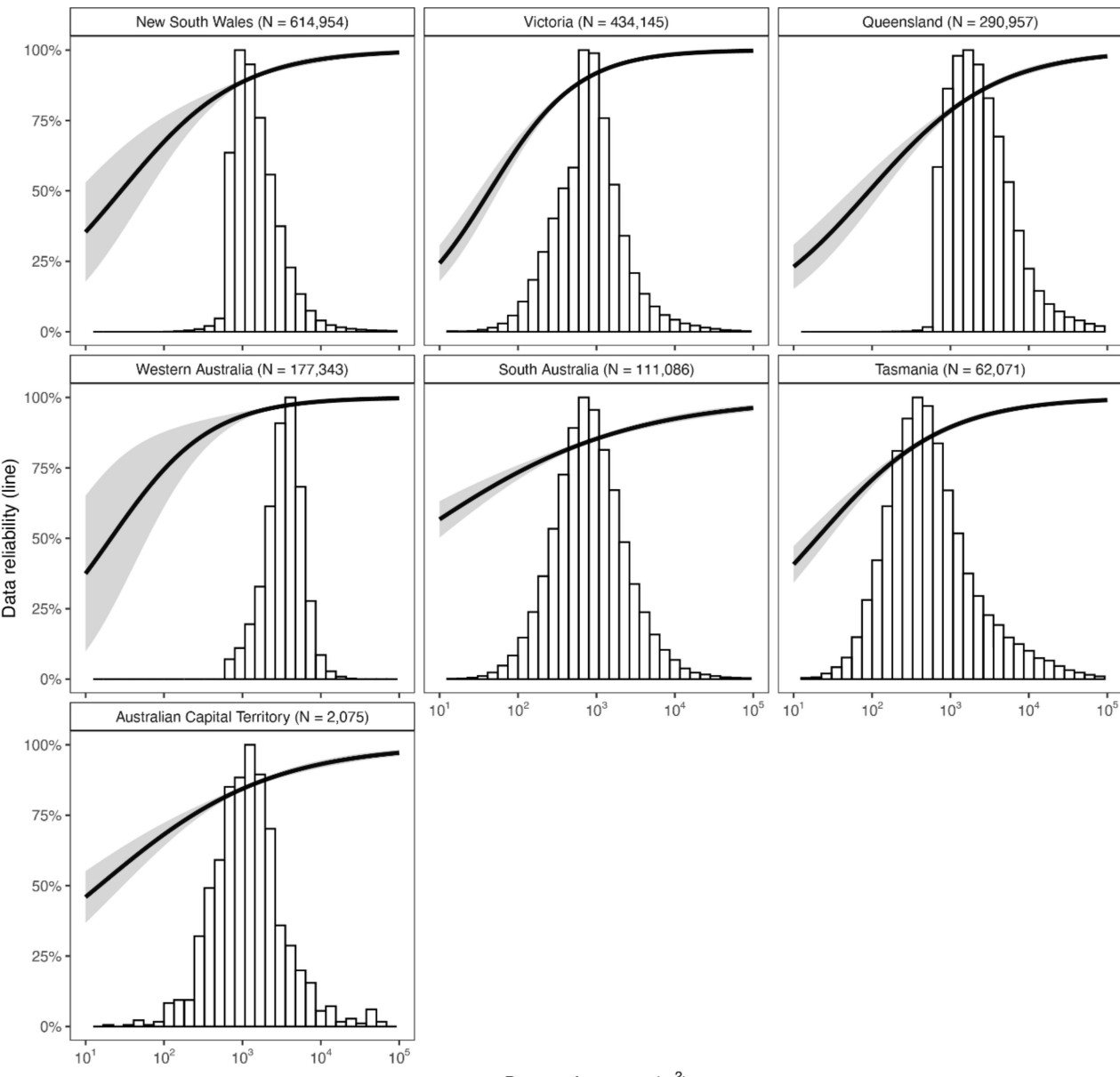

**Figure 3.** Dam detection reliability as a function of geographic region and dam surface area (m²). Histograms and x-axes represent the distribution of all documented dam sizes, while faceting represent States and Territories in Australia (with sample size reported in the facet titles). Lines (±95% C.I) indicate the probability of a reliable entry extracted from the best-fitting generalised linear model following Akaike Information Criterion. Low probabilities indicate high frequencies of wrongly classified dams (false positives), whereas high probabilities indicate high frequencies of dams correctly documented (true positives). We omitted data for the Northern Territories because there were too few documented dams to carry out a formal probability assessment.

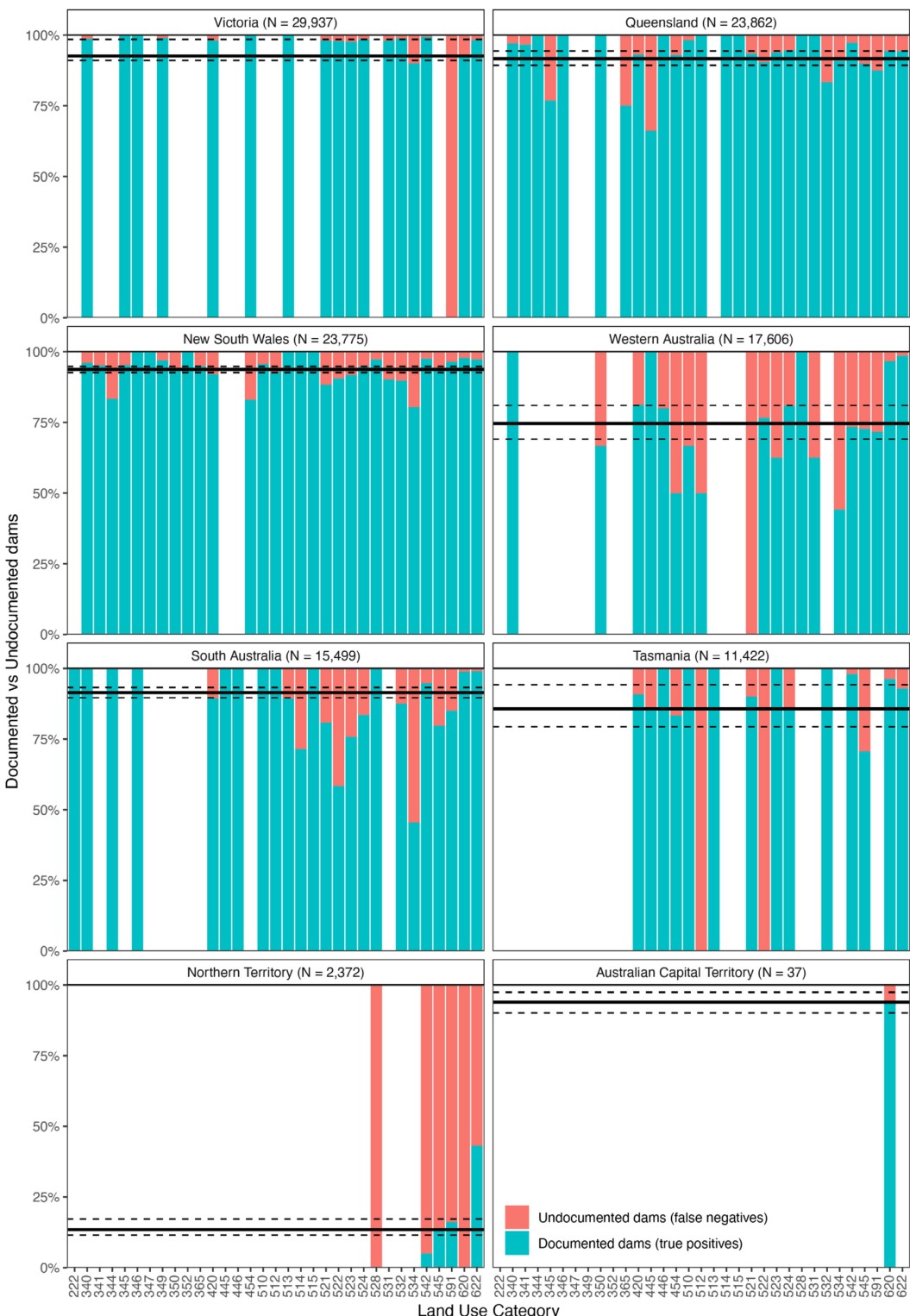

**Figure 4.** Frequency of undocumented farm dams (false negatives) encountered in the 33 agricultural land types with the highest reported dam densities (>2 dams km$^{-2}$) across Australia (see Figure S2 for the names of the land use categories used in this analysis). We searched for dams by randomly sampling sites among these land use types. The number of searched sites in each region is reported in the facet title. Horizontal lines indicate overall percentages of detected dams across all land use types over the total (±bootstrapped 95% confidence intervals). Missing columns indicate land use categories that are absent from the State or Territory.

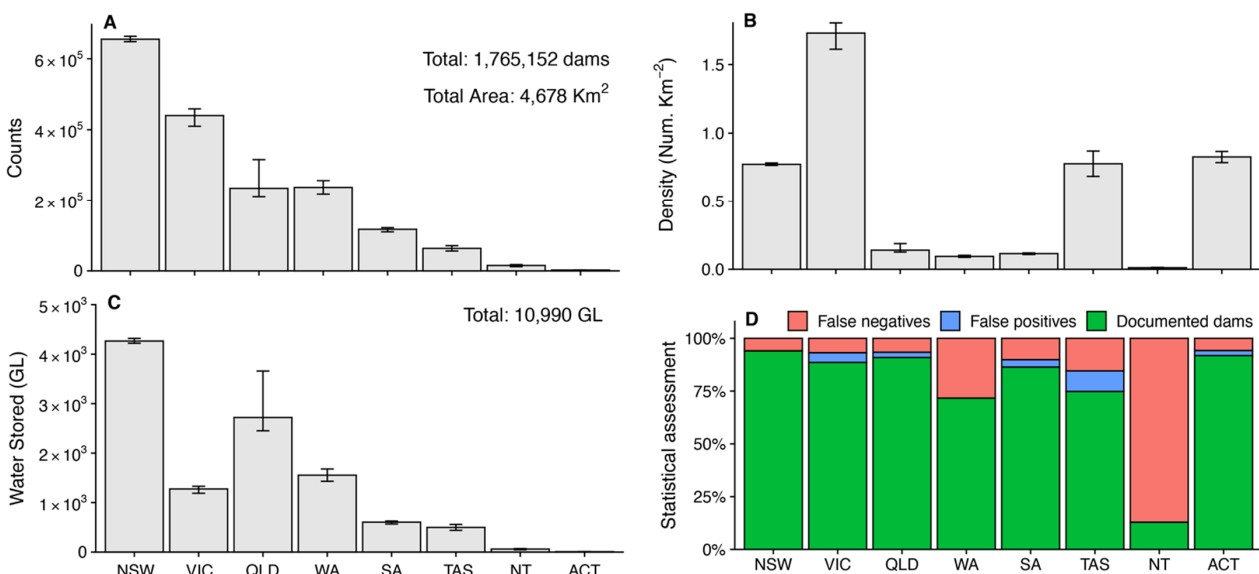

**Figure 5.** Final statistics for Australian farm dams in each State and Territory. (**A**) Total counts, (**B**) overall densities and (**C**) cumulative water capacities (GL) are calculated after removing unreliable entries (false positives) and adding expected undocumented dams (false negatives). The size and water capacity of undocumented dams were calculated based on manual tracing (Figure 2). Grey bars indicate medians, while error bars represent the bootstrapped 95% confidence intervals. (**D**) Relative contributions of false positives and false negatives compared to the total documented dams in each State and Territory.

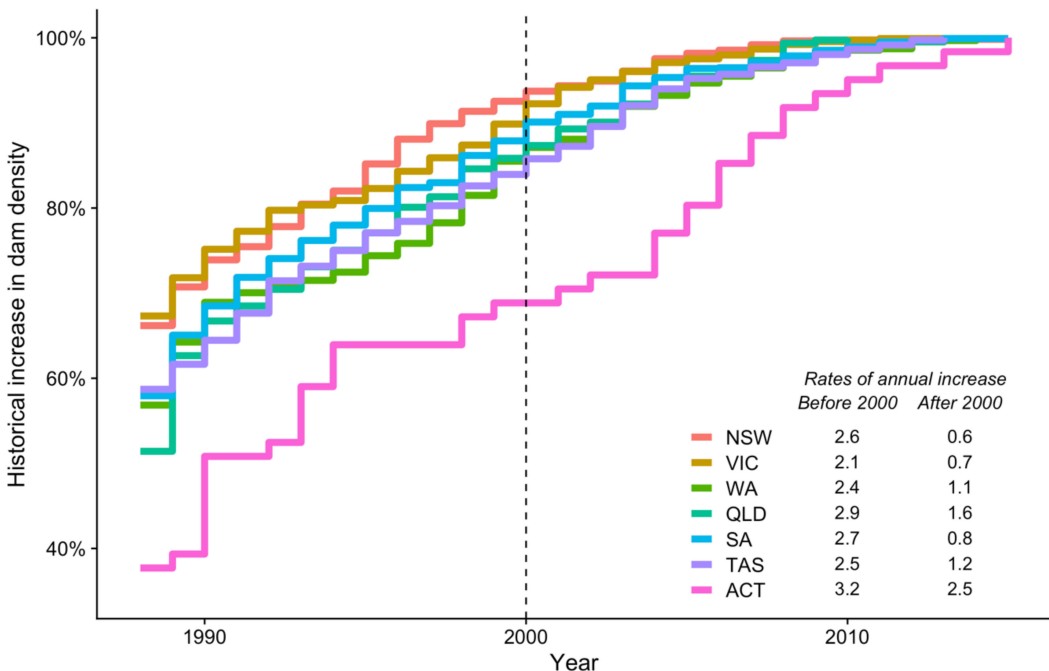

**Figure 6.** Historical trends in dam density between the years 1988 and 2015 in each State and Territory of Australia. The embedded table shows annual rates of proportional increases in dam densities, both before 2000 and after 2000 (dashed line). There were too few data to calculate historical rates for dams in the Northern Territory. See Figure S6 for absolute and relative rates of annual increase for each region and Figure S7 for trends in farm dam size over time.

## 4. Discussion

In 2018/2019 there were 1,765,152 farm dams in Australia, of which 202,119 (11% of the total) remained undocumented. Freshwater provides an essential service to Australia's economy, with the total annual value of irrigated agriculture last estimated at AU$17.7 billion, mostly in Victoria ($4.9 billion), Queensland ($4.5 billion) and New South Wales ($4.4 billion) [45]. There are 9968 GL of water currently used in Australian agriculture, of which 13.3% comes from farm dams or tanks (1324 GL) [46]. Using the percentages of undocumented farm dams calculated here for Australia (12.5%), we can approximate that on average undocumented farm dams are associated with (AU$17.7 B $\times$ 13.3% $\times$ 12.5% =) AU$294 million of Australia's revenue. This back-of-the-envelope calculation indicates that ensuring appropriate monitoring and management of farm dams across the country is likely to have important economic rewards, other than ecological and environmental benefits.

Perhaps the most important reason for increasing investments into monitoring farm dams is water security [47]. We estimated that farm dams in Australia hold 10,990 GL and we showed where these freshwater reserves are. Trends in available freshwater are becoming of increasing concern under anthropogenic climate change. Climate change is reducing rainfall, increasing evaporation and intensifying extreme weather events and droughts [48]. In some area of Australia, growing-season rainfall have already declined by 14–20% since the 1990s [49,50]. Population growth will nearly double worldwide food consumption by 2050, and current water availabilities in Australia could fail to meet future demands [18]. We would therefore expect an overall reduction in the available water in farm dams, but there is no data to test this prediction. Hence, a promising next step would be to complement our study with satellite tools to track interannual trends in water availability within farm dams.

Investing now in better monitoring techniques for farm dams is most cost-effective than ever. Specifically, we here found a monotonic decline in historical rates of dam development among Australian States and Territories: from 2–3.4% before 2000, to 0.5–1.5% after 2000, to 0.05–0.8% after 2010. This decline is consistent with the 2003 Farm Dams Act that limited construction rates in South-East Australia [51]. However, we are unaware of any other policy intervention or natural event that could explain this nation-wide plateau, possibly indicating saturation of available space or farm dam demand—although we also found that on average modern farm dams are larger than older ones. Regardless of the underlying drivers, if this trend continues, dam numbers will nearly stabilise before 2030, which means investing now into a national farm dam database would require fewer updates than in the past.

There are several ways in which this work can support new research. For example, farm dams have unique properties that make them a hotspot for methane emissions—a greenhouse gas that is 34 times more potent than carbon dioxide [9,10,52]. Given Australia's commitment to substantially reduce emissions by 2050, the contributions of farm dams to climate change must be monitored and regulated—as recommended by the 2019 Refinement of IPCC Guidelines [53]. The dataset presented here on size and location of farm dams can help government agencies (e.g., Dept. of Agriculture, Water and the Environment) to include their greenhouse gas emissions in the Australian National Greenhouse Gas Inventory. As another example, our data can help manage biological invasions. In arid habitats, farm dams provide a refuge that pests can use as stepping-stones to spread across the country (e.g., the cane toad *Rhinella marina* in north Australia; Letnic et al., 2015) [54]. Knowing where farm dams are can, therefore, inform on invasion fronts. Moreover, our map of farm dams could help to predict species richness and distribution across Australia [4,5], manage water quality [55,56], nutrient leaching [57] or sediment delivery [58,59].

## 5. Conclusions

Human practices have created millions of artificial water bodies in rural areas across Australia, but we still lack basic information on their impacts on the environment. To encourage future research on sustainable management of Australian farm dams, we created a

free interactive website (www.AusDams.org) to share our results and analyses with the Government, farmers, scientists and the general community (Figure 7). We designed this portal to ensure maximum simplicity: the user only needs to navigate on a map to any area of Australia to generate tailored statistics, plots and tables on various aspects of farm dams (e.g., count, density, total surface area, size distribution, water capacity). Moreover, we incorporated in our farm dam statistics both false positives and false negatives, which can be important to inform where to prioritise new mapping efforts (Figure 7). Specifically, policymakers can decide to focus on the region with the highest number of omitted farm dams (Western Australia), or with the most significant percentage of omitted farm dams (Northern Territory), or with the highest overall agricultural value (New South Wales), or with the highest cost of water (Northern Territory). Our portal can also facilitate managing licenses or help choosing the location for new dams. To support all these applications, we are committed to keeping expanding the data in AusDam.org as they become available.

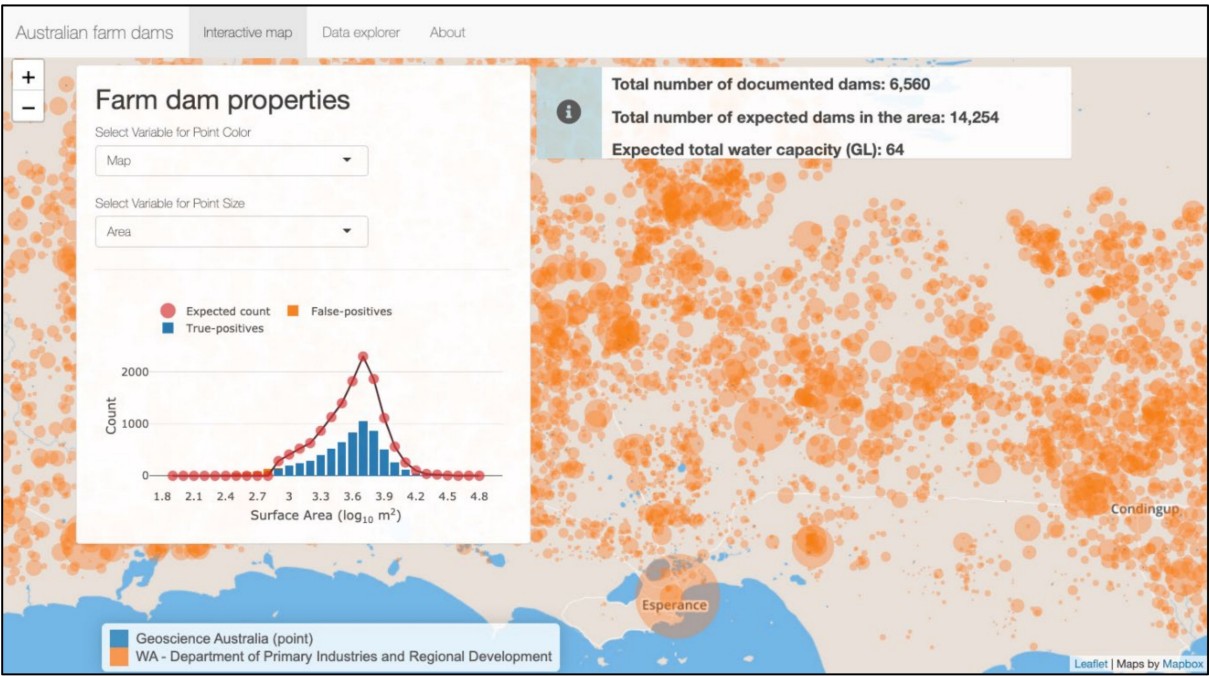

**Figure 7.** The online guided user interface of AusDam.org. This example shows all dams (red dots) in the region near Esperance in Western Australia. The panel on the left allows the user to choose the variables to represent as point colour (e.g., the region of the map, data source) and as point size (e.g., surface area, perimeter, estimated water capacity) for each dam. The histogram shows expected counts (dots), documented dams (blue bars), and the expected false positives (orange bars), following all calculations presented in this study. The banner at the top summarises the overall statistics. The two tabs at the top presents the raw data (Data explorer) and information about the project and the methods (About).

**Supplementary Materials:** The following items are available online at https://www.mdpi.com/2072-4292/13/2/319/s1. Figure S1: Confusion matrix for farm dam detection with our deep learning CNN. Figure S2: The surface area of unreported dams used to estimate their median water capacity. Figure S3: Examples of undocumented farm dams that were identified with our deep learning CNN. Figure S4: Step-by-step graphical diagram for the methods to calculate the absolute and relative rates in the construction of Australian farm dams. Figure S5: Predicted probability for a successful verification (true positive) for each dam in our dataset. Figure S6: Estimated rates of annual increase in dam densities between the years 1988 and 2015 in each State and Territory of Australia. Figure S7: Historical trends in farm dam size with year of construction for each State and Territory. Table S1: Summary table for all farm dam datasets used in this study.

**Author Contributions:** Conceptualisation, M.E.M., N.W. and P.I.M.; methodology, validation, formal analysis and data curation M.E.M. and N.W.; investigation and resources, M.E.M., N.W. and P.I.M.;

writing—original draft preparation, M.E.M.; writing—review and editing, M.E.M., N.W. and P.I.M.; visualisation, M.E.M.; project administration, M.E.M.; funding acquisition, M.E.M. All authors have read and agreed to the published version of the manuscript.

**Funding:** This research was funded by the Alfred Deakin Postgraduate Research Fellowship by Deakin University.

**Institutional Review Board Statement:** Not applicable.

**Informed Consent Statement:** Not applicable.

**Data Availability Statement:** Publicly available datasets were analyzed in this study. See Table S1 for more information.

**Acknowledgments:** This work was supported by the Alfred Deakin Fellowship scheme and the Western Australia Department of Primary Industries and Regional Development (WA DPIRD). We thank Tertius de Kluyver and Shanti Reddy (Dept. of Industry, Science, Energy and Resources), Richard George (WA DPIRD), Meredith Holgerson (Cornell Uni.), Luis Torres ($A^2I^2$, Deakin Uni.) and Erik Schmidt (Uni. Southern Queensland) for helpful comments and suggestions. We also thank Tingbao Xu and Michael Hutchinson (Australian National Uni.), Alistair Grinham (Uni. of Queensland), Quinn Ollivier (Deakin Uni.), Bryce Graham and Henry Maxwell (Water and Marine Resources Division, Tasmania) and Rebecca Lett (Department of Environment, Land, Water and Planning) for support during data collection.

**Conflicts of Interest:** The authors declare no conflict of interest.

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
