# Peer review of "A Continental-Scale Assessment of Density, Size, Distribution and Historical Trends of Farm Dams Using Deep Learning Convolutional Neural Networks"

_remotesensing, doi:10.3390/rs13020319_

Round 1
Reviewer 1 Report
In this research, the authors made an interesting remote sensing investigation to figure out the quantity of farm dams in Australia. They found that there were about 11% farm dams were undetected in the past datasets. The historical trends of dam density could offer detailed information to managers. However, the paper is poorly organized and lacks important clarities in methods and results.
- The detailed description of the existing farm dams data should be made. How were they been yielded? What’s the temporal and spatial coverage of the current datasets? How about their reported accuracy?
- The extracted farm dams were not shown. A figure of the extracted farm dams would help the readers to directly judge the performance of your algorithm.
- The remote sensing data source were not clearly explained. In the article, remote sensing image with different spatial resolution were used. When training the deep learning model, images with 0.45m and lower spatial resolution (1-5m) and only RGB bands were used. However, in historical trends analysis, Landsat 5 and Landsat 7 images were used. The spatial resolution ratio of Landsat visible bands and the previous high resolution images is around 30. Such huge spatial resolution ratio is a big challenge for any remote sensing image processing algorithms. The affect of different spatial resolution to the farm dam detection must be discussed. What’s more, the spectral bands of Landsat image that been used in the detection algorithm should also be clarified.
- About the water volume calculation. The authors used two models developed in the previous studies. Two problems must be mentioned here. Firstly, the number of samples of the two studies are very small compared with the current manuscript. Their models were built based on 73 samples and 66 samples, respectively. What’s the dams’ area in their researches? Domains of the simple empirical models are important in applying the models into different sample sets. Could the models been directly used in this research? In addition, the two models yield quite different output volumes. Like the authors plotted in Figure 2, 1000m² was the mode of dam areas in many regions of Australia. Under this condition, the dam volume difference that yielded from the two models is about 30%. The simple average of the two models was not credible.
- In the result, the total farm dams and their historical trends were shown. It is clear that the historical trends were generated from Landsat image series from 1988 to 2015. However, the temporal range of total dams was not clear. Was it year 2015? Was the dams’ quantity comparable in section 3.1.4 and 3.1.6?
- In figure 3, the authors plotted the frequency in 33 agricultural types. Which 33 types? They never mentioned that in the rest of the article.
- There are so many small mistakes and typos in this article. Thus, I suggest a thorough proofreading of the manuscript and editing. Below are some specific comments:
Page 2, line 72. “3.1” should be “2.1”. All the subtitles in section 2 need to be rectified.
Page 3, line 126. “false positives or commission”. I think it should be “false positives or omission”.
Page 4, line 161. “3.1 subsections”. Normally, we do not name a subsection like this.
Reviewer 2 Report
Summary:
This manuscript describes a protocol combining remote sensing and deep-learning that has been developed by the authors to map farm dams in Australia, and to complement the official records of farm dams. The protocol enables to monitor the water surface area, reservoir storage, and density of farm dams since 1988.
Broad comments:
The paper is clearly written, and the methodology and results are well explained. I am not an expert in artificial intelligence and I find useful the clear and detailed description of the protocol developed by the authors. While the approach is not novel and combines existing tools and methods (algorithm to detect water from satellite images, fastai library for the deep learning detection model) the results are still interesting for the Australian context. It also seems to me that the methodology could be applied to other parts of the world if enough information on existing farm dams is available to feed the artificial intelligence algorithm. Moreover, the internet platform http://ausdams.org/ that has been designed to present the results to a broad audience from local farmers to scientists seems like a valuable tool for any stakeholders that would like to access the data and results.
Specific comments:
1) The authors claim that they have developed the first continental-scale assessment on density, distribution and historical trends of farm dams (e.g. l. 17, l. 64). However, many methods have already been developed to monitor small and large reservoirs in many parts of the world. I would appreciate at least a brief summary in the introduction to review the literature on existing water detection methods and artificial intelligence approaches, and on the benefits and novelty of the manuscript approach compared to what has already been done.
2) Two different storage-area relationships are used to assess the volume of stored water is farm dam reservoirs (l. 88). The difference can be large for the largest farm dams (> 105 m2). Did you include this uncertainty when assessing the water stored in reservoirs (section 3.1.5)? If not, is it possible to provide an average estimate of such uncertainty?
3) One of the main results of the paper is the decline in historical rates of dam development. However, I did not find any information in the manuscript on the evolution of the average size of farm dam reservoirs. Is there a possibility that less dams have been built after 2000 but that the ones that have been built are larger than before 2000? Can you please provide information on the evolution of the size of dams and stored water in the results and discussion?
Round 2
Reviewer 1 Report
Authors addressed all the comments and revised the manuscript.